# Pathogenesis of Dementia

**DOI:** 10.3390/ijms24010543

**Published:** 2022-12-29

**Authors:** Janusz Wiesław Błaszczyk

**Affiliations:** Jerzy Kukuczka Academy of Physical Education, 72 Mikołowska Street, 40-065 Katowice, Poland; j.blaszczyk@awf.katowice.pl

**Keywords:** dementia, Alzheimer’s disease, brain energy metabolism, cognitive metabolism, brain aging, neurodegenerative disorders

## Abstract

According to Alzheimer’s Disease International, 55 million people worldwide are living with dementia. Dementia is a disorder that manifests as a set of related symptoms, which usually result from the brain being damaged by injury or disease. The symptoms involve progressive impairments in memory, thinking, and behavior, usually accompanied by emotional problems, difficulties with language, and decreased motivation. The most common variant of dementia is Alzheimer’s disease with symptoms dominated by cognitive disorders, particularly memory loss, impaired personality, and judgmental disorders. So far, all attempts to treat dementias by removing their symptoms rather than their causes have failed. Therefore, in the presented narrative review, I will attempt to explain the etiology of dementia and Alzheimer’s disease from the perspective of energy and cognitive metabolism dysfunction in an aging brain. I hope that this perspective, though perhaps too simplified, will bring us closer to the essence of aging-related neurodegenerative disorders and will soon allow us to develop new preventive/therapeutic strategies in our struggle with dementia, Alzheimer’s disease, and Parkinson’s disease.

## 1. Introduction

Consciousness is a physiological state of functioning of the human brain that allows it to receive and interpret signals from the environment. These signals are written into the functional structure of the brain, and their permanent memory traces are used to plan and implement behaviors necessary for survival. Learning and memory, which are the basis of human consciousness, are continuous processes that last throughout life, and their effectiveness depends in turn on the efficiency of many physiological processes carried out at the cellular, tissue, and systemic level. Disruption of each physiological cycle leads, in the long or short term, to the impairment of learning and memory and the disintegration of human consciousness collectively referred to as dementia. Such sequences of pathological phenomena seem to be a natural consequence of the aging processes of the brain and the whole organism. However, their dynamics depend on the weakest link of cognitive processes, which, in the case of their complexity, makes it difficult for scientists to pinpoint the unambiguous cause of dementia and Alzheimer’s disease. In the current attempt to bring closer the problem of diagnosis and possible prevention of dementia, I focused on the analysis of the current knowledge about brain dysfunction in the aging process. The knowledge gathered so far allows us to indicate several pathogenic phenomena leading to the development of dementia. Three of them seem to have a leading role in disturbances of consciousness. The main one is the age-related impairment of energy metabolism at the cellular level. It leads to gradually increasing metabolic disorders and disturbance of homeostasis, which is vital for brain activity. They result in several behavioral dysfunctions accompanying old age, such as limited physical and cognitive activity, and sleep disorders. These dysfunctions further exacerbate the pathophysiological changes in the brain, leading to the massive death of nerve cells. This rapidly growing process of self-destruction of the aging organism leads to dementia, i.e., inhibition of interaction with the environment, which is the meaning of life.

## 2. The Leading Role of Energy Dysmetabolism in Dementia Development

The human brain has extraordinary cognitive abilities. This is due in particular to the enlarged neocortex and its unusual susceptibility to plastic changes. This allowed for the development of such human abilities as sentience, consciousness, perception, memory, speech, and abstract thinking [1]. These abilities result primarily from the integrative and perceptual function of the prefrontal cortex, which interacts with other areas of the neocortex, the hippocampus, and centers processing visceral information [1]. Pyramid neurons with a very complex morphology play a special role in the formation of neural networks. Each pyramidal neuron can form up to several thousand synaptic connections, allowing for the coding of enormous amounts of information. Although the basic anatomical structure of the cerebral cortex is genetically determined, the process of synaptic networking begins in the postnatal period. The inflow of sensory signals causes pyramidal neurons to enter the phase of modeling the structure of synaptic connections, by creating new and removing redundant connections [1]. The process of intensive shaping of the cerebral cortex begins in the second year of life and is characterized by an extraordinary increase in cognitive abilities [1]. The high intensity of cognitive processes depends on the influx of perceptual streams, their novelty, intensity, and emotional–motivational value which shape the structure of nervous networks accordingly. Cognitive processes, although with different dynamics, shape individual personality until adulthood. They are also responsible for the creation of an egocentric representation of our environment that allows for optimal adaptation to it.

The brain is a highly energy-demanding organ whose functioning depends firstly on a stable and efficient energy supply. In the waking state, the human brain, constituting only 2% of the total body weight, consumes, under physiological conditions, up to 20% of oxygen and 25% circulating glucose [2]. The human brain consumes about 90 g of glucose per day, and the major energy demands of the brain are associated with neuronal signaling and account for ~70% [3]. Oxidation of fatty acids and ketone bodies only marginally replaces glucose as a cerebral energy source. The brain cannot take free fatty acids up from the blood and must rely on producing fatty acids from carbohydrates in astrocytes. Consequently, all tissues except the brain may switch from glucose to fatty acids, thus theoretically saving glucose for the brain [4]. Unfortunately, an increased level of fatty acids in the blood due to a fatty diet inhibits glucose metabolism in the liver, which is called the “glucose-sparing” effect [5]. The effect contributes greatly to the development of type 2 diabetes and upregulates glucose to a toxic level [6].

Central nervous system (CNS) complications induced by toxic levels of glucose are multifactorial and are relatively little understood. It is now evident that malfunctioning of the blood–brain barrier may play a significant role in diabetes-dependent CNS disorders. Chronic exposure of cells and tissues to high glucose concentrations results in the non-enzymatic glycation of proteins that elevates the production of reactive oxygen species (ROS) [7]. A combination of hyperglycemia, protein glycation, and oxidative stress results in the massive synthesis of gluconic acid, a very toxic product of glucose oxidation. This organic compound is produced intensively in aging brain tissue [7]. Gluconate as a potent iron and copper ions chelator can affect energy metabolism, firstly by decreasing oxygen transport by hemoglobin and in the aging brain, attenuating or ceasing the mitochondrial electron transport chain both in neuronal cells and oligodendrocytes, thus initiating their apoptosis [7]. Chronic exposure, even to low levels of iron and copper chelator, may potentiate the effect of nicotinamide adenine dinucleotide (NAD) deficiency and escalate the aging dysmetabolism in the brain [8,9,10].

The cerebral cortex comprises, by volume, about 80–85% of the adult human brain. It has been estimated that the human cerebral cortex may be composed of 10–20 billion neurons, each of which may be synaptically connected with several thousand other neurons [11]. With each new experience and each remembered event or fact, the brain slightly rewires its physical structure [12,13,14,15]. The neocortex accounts for 44% of the brain’s energy consumption [11]. Maintaining resting potentials in neurons and glia accounts for 28% and 10% of energy consumption, respectively. Recovery of resting potential accounts for an additional 13%. Calcium movements associated with transmitter release and transmitter recycling each account for less than 1% [11]. Consequently, assuming that the average pyramidal neuron makes tens of thousands of synaptic connections, the cost of a single action potential is so high, that fewer than 1% of pyramidal neurons can be active concurrently [11].

### 2.1. Nicotinamide Adenine Dinucleotide and Brain Aging

All neurochemical processes are energy-dependent, and a growing body of evidence suggests that brain energy-related dysmetabolism can be considered a root cause of brain involution. Dysmetabolism may disrupt all biochemical processes in the brain, both at the cellular and tissue levels. Currently, it seems that energy metabolism, sensory information flows, and adult neurogenesis is essential for the networking of limbic areas and their stability [16]. The energy metabolism of neural networks relies exclusively on oxidative phosphorylation with oxygen and glucose being the main substrates for the process [17].

The kinetics of neuronal respiration is limited by an uninterrupted cycle of reduction/oxidation of nicotinamide adenine dinucleotide (NAD).NAD is an essential pyridine nucleotide that serves as a cofactor and substrate for critical cellular processes involved in oxidative phosphorylation and ATP production, DNA repair, epigenetically modulated gene expression, intracellular calcium signaling, and immunological functions [8,9,10,18,19]. Consequently, NAD level is the rate-limiting factor in oxidative phosphorylation. The cellular NAD levels, and therefore the efficiency of energy metabolism, decline in senescence neurons. NAD levels decline at the cellular, tissue/organ, and organismal levels during aging. Age-related NAD decline leads to mitochondrial dysfunction and metabolic abnormalities [8,9,20]. In physiological conditions, NAD is continually turned over by three classes of NAD-consuming enzymes: the NADases (CD38, SARM1), the protein deacetylase family of sirtuins, and PARPs [8,9]. Both CD38 and PARP1 are NAD-dependent enzymes that significantly contribute to cellular NAD degradation [9].PARPs are involved in the process of DNA damage repair. Their excessive activation, however, may lead to overconsumption of cellular NAD, and consequently metabolic collapse, and cell death. It is well-documented that excessive PARP activity is symptomatic for the aging brain [20]. NAD deficiency, due to increasing demands for NAD-consuming enzymes as occurs in the aging process of the brain, causes the mitochondria to become less efficient and neurons cannot produce enough energy, especially ATP [21,22]. In old age, the NAD deficit can reach up to 20% of physiological needs [8,9].

Lactate is considered a potential energy substrate capable of supporting neuronal activity in conditions of reduced oxygen supply, e.g., during exercise [23]. Almost 10% of glucose entering the brain yields lactate in a metabolic process of aerobic glycolysis [24]. However, aerobic glycolysis and lactate production is a metabolic process observed only in astrocytes but not in neurons [24]. Astrocytes are unique in that they use glycolysis to produce lactate, which is then shuttled into neurons. Lactate is now recognized as having a central role in neuronal functioning altering intracellular NADH/NAD ratio [3].

### 2.2. Neural Control of Energy Metabolism

During severe energy crises, glucose can be produced in the process of gluconeogenesis [25], a rescue physiological process that allows maintaining indispensable blood glucose concentration in conditions of its long-lasting deficiency. In humans, the process occurs during periods of prolonged fasting, starvation, low-carbohydrate diets, and even intense exercise load [25]. Recent studies have demonstrated the existence of gluconeogenesis in astrocytes, which is recognized as an important alternative lactate and glucose source for neurons, specifically in ischemic conditions [25].

The metabolism of the cognitive neuronal networks consists of multiple interconnected metabolic pathways that may happen efficiently and at the required rate only in the state of steady internal, physical, and chemical conditions, including reactants’ concentration, tissue temperature, and energy supply. Energy metabolism is a core set of catabolic pathways that allows the maintenance of the resting cellular homeostasis and its recovery after neuronal activity. Additional energy must be allocated for extracellular metabolic processes including waste and neuronal debris removal, allowing recovery of brain tissue homeostasis. From this perspective, disturbance in any metabolic process may result in the rapid or postponed development of chronic brain dysfunction.

The autonomic nervous system (ANS) plays a key role in the control of energy balance and glucose homeostasis [26,27,28]. Interoceptive information such as blood glucose levels, blood circulation, oxygenation, osmolarity, as well heart rate, and respiratory rhythm are sent via the parasympathetic nervous system to the insular cortex through the brainstem. In response, the brain adjusts the desired states of visceral organs and physiological processes allowing for energy metabolism optimization [26,28]. For instance, low glucose levels generate interoceptive signals that activate the feeling of hunger and trigger eating behaviors [29].

The hypothalamus is the central controller of homeostasis that regulates basic processes such as food and water intake, energy expenditure, responses to stress, blood pressure, and all processes that are required for survival [30]. The nuclei of the lateral hypothalamus are directly connected to the motivational–emotional system, playing an important role in controlling reward and motivation. The orexin neurons of the lateral hypothalamus form direct synaptic connections with the limbic emotional system, especially with the nucleus accumbens, the ventral tegmental area, and the amygdala, playing an important role in controlling reward and motivation. Hypothalamus is also an important integrator and controller of the autonomic, endocrine, somatic, and motivational–emotional functions that are essential for maintaining homeostasis, the stable internal environment of the body [30,31]. To perform these functions effectively, the hypothalamus monitors both the environment of the brain and the whole organism sensing the blood composition, especially the content of some neurohormones, peptides, fatty acids, and most important for the brain, glucose, and lactate.

Within a few regions of the brain including the hypothalamus, the blood–brain barrier is not complete, which allows insulin, IGF-1, ghrelin, leptin, other peptides, and even fatty acids to penetrate the barrier and gives a more precise sense of the nutritional value of the food consumed. Unfortunately, the incomplete blood–brain barrier increases the susceptibility of the hypothalamic nuclei to focal damage by free fatty acids and consequent impairment of homeostatic and metabolic control [32,33]. Metabolic inflammation in the mediobasal hypothalamus is marked by the activation of astrocytes and microglia [34,35,36]. The resultant metabolic syndrome is characterized by uncontrollable weight gain accompanied by multiple endocrine abnormalities, and psychoneurological symptoms such as attention deficit, memory impairment, and reduced impulsive control accompanying hypothalamic deficiency [33].

The respiratory center is located in the brainstem, in the medulla oblongata, and the pons. The center is responsible for generating and maintaining the rhythm of respiration and also adjusting this in homeostatic response to physiological changes [37]. The respiratory center receives input from chemoreceptors, and the hypothalamus to regulate the rate and depth of breathing. The central chemoreceptors located in the medulla oblongata are primarily sensitive to changes in blood pH and partial pressure carbon dioxide. The carotid bodies, located in the bifurcation of the common carotid arteries, detect changes in the composition of arterial blood, mainly changes in the partial pressure of oxygen and carbon dioxide, blood pH, and temperature.

Changes in the blood acid-base balance result from changes in metabolism and breathing. Excess carbon dioxide causes the pH of blood to decrease and its intracellular levels to increase. The increasing level of carbon dioxide increases glycolysis via the phosphorylation of glucose, and glucose 6-phosphate is produced. This leads to a reduction in intracellular phosphate and a consequent shift of phosphate into cells. Serum phosphate falls within 20 min of hyperventilation, and it persists for 90 min after the ventilation returns to normal [38,39]. The intracellular depletion of phosphate leads to a decrease in ATP production and thus deficient energy metabolism limiting neuronal activity. This causes the generation of ATP from ADP and a consequent increase in AMP, which, via 5’ AMP-activated protein kinase (AMPK), stimulates glucose uptake. However, with phosphate deficiency, glucose is anchored in neuronal cells and is used for energy production [17].

The control of glucose metabolism can be divided into two processes. First is insulin-dependent glucose storage in skeletal muscle, liver, and white adipose tissue [40]. The liver is crucial for the maintenance of normal glucose homeostasis—it stores glucose postprandially and produces glucose during fasting or starvation [41,42]. In the fasted state, the liver provides glucose to maintain euglycemia and fuels the brain. Hepatic glucose release accounts for almost 90% of endogenous glucose production [42].Key regulators of hepatic glucose metabolism act through diverse mechanisms including the balance of hormones such as insulin, glucagon, catecholamines, corticosteroids, and growth hormone [41,42]. The result of all insulin actions is a fast stabilization in blood glucose levels. Recent studies demonstrated also that insulin stimulates bone cells (osteocytes) to release osteocalcin, an endocrine hormone that adjusts the glucose metabolism to calcium/phosphate metabolism controlled in a bone resorption-dependent manner [40].

### 2.3. Bone-Derived Phosphate Determines Brain Aging and Neurodegeneration

Besides oxygen, glucose and phosphate play an essential role in cognitive metabolism [39,43]. At the cellular level, metabolic crises may be caused by oxygen, glucose, and phosphate deficiency mainly. In response to a chronic deficiency of oxygen, neuronal metabolism declines, and simultaneously, the hypoxia-inducible factor (HIF-1α) tries to restore oxygen homeostasis [44]. In physiological conditions, HIF-1α undergoes rapid degradation, while hypoxia blocks degradation leading to the accumulation-dependent activation of over 100 genes that are involved in crucial aspects of brain physiology, including neuronal cell survival, glucose metabolism, and angiogenesis attempting to reduce negative effects of hypoxia [45]. In the case of failure of the compensatory mechanisms, the neurons enter apoptosis.

The human body turns over the body weight equivalent in ATP daily [46]. The ATP-derived phosphate is used then in multiple phosphorylation cycles, including glucose phosphorylation, which all contribute to the regulation of vital neuronal processes necessary for brain activity and metabolism. For example, protein phosphorylation regulates interactions between components of neuron-neuron and neuron-glia synergies. Importantly, each pyramidal neuron in cortical networks, to maintain resting membrane potential and to fire an action potential, consumes almost three times more ATP than other neurons [11]. To maintain neuronal homeostasis, a stable level of phosphate must be provided. The extracellular phosphate supply is essential for all phosphorylation processes necessary for modeling and maintaining the dynamic structure of cortical networks. Recent data allows proposing a hypothesis that bone-derived phosphate plays a vital role as an energetic booster in neurons. In particular, in young, healthy individuals, osteocytes compose 90–95% of all bone cells and maintain bone homeostasis, integrity, and functional remodeling. Importantly, the osteocytes also orchestrate bone remodeling with motor activity [47,48]. The activity of osteocytes is controlled, among others, by calcitonin, parathyroid hormone, growth hormone, and vitamin D. Such multifactorial control allows for maintaining calcium and phosphate homeostasis. Osteocyte viability allows for maintaining homeostasis for decades up to old age. As we age, the cells die resulting in reduced bone remodeling and dysregulation of energy metabolism.

Phosphate supply to the brain is controlled in a neuronal activity-dependent manner, making phosphate the brain metabolic booster. Serum phosphate homeostasis is maintained through a complex interaction between intestinal phosphate absorption, renal phosphate handling, and the cellular intake of phosphate. The homeostasis is under the direct hormonal influence of calcitriol, PTH, and phosphatonins, including fibroblast growth factor 23 (FGF-23) [49]. FGF-23 is released by osteocytes to increase the excretion of phosphate in the urine. A transient shift of phosphate into the active cells is stimulated by insulin or IGF-1, glucose, and respiratory alkalosis. The main source of phosphate is bone reserves (85%) whose resources decline with age [49]. Due to limited access to phosphate, increased glucose blood levels may provoke hypophosphatemia. A combination of hypophosphatemia with hyperventilation may result in a greater decline in serum phosphate due to the rapid uptake of phosphate and glucose into neurons [38,39]. In the case of uncontrolled hypophosphatemia, cellular metabolism in the brain and respiratory muscles may fall rapidly, causing patients’ death [38,39]. Concluding, the aging-related collapse of osteoblast-controlled calcium/phosphate homeostasis may act like the mythological Atropos cutting the thread of human life.

## 3. Systemic Dysmetabolism

Human energy metabolism is supplemented by several hormonally controlled processes. The integrated energetic processes are controlled by the growth (GH) and thyroid (T4/T3) hormones. T3 acts on most active cells such as neurons and muscle cells, increasing the production of Na/K-ATPase which serves as an ion pump creating an electrochemical gradient across the plasma membrane [50]. In the brain, the thyroid hormone elevates the number of mature oligodendrocytes and directly enhances the expression of myelin-specific genes. Additionally, by increasing the rate of glycogen breakdown and gluconeogenesis, T3 affects almost every physiological process in the body including growth and development, metabolism, body temperature, and heart rate.

The secretion of growth hormone (GH) in the pituitary is regulated by the neurosecretory nuclei of the hypothalamus. The growth hormone is secreted in a pulsating manner and the largest GH peak occurs about an hour after the onset of sleep [51]. The released GH stimulates primarily the liver synthesis of insulin-like growth factor(IGF-1), that in turn increases the levels of glucose and free fatty acids. It plays a particularly important role in child growth and development but has also anabolic effects in adults [52,53]. Dysregulation of IGF-1 activities may induce growth disorder, and many types of age-related diseases including diabetes, neurodegenerative diseases, cancers, arteriosclerosis, and osteoporosis.

Realization of time-consuming processes such as neuroplastic changes in cortical networks requires the supportive activity of IGF-1, which shares common receptors in the brain with insulin [51,53,54]. Their final signaling effects, however, are quite different due to their quite different half-life. The half-life of insulin ranges from about 10–15 min whereas the IGFs’ may exceed 15 h [51]. Insulin shows short-lasting effects contributing to glucose storage in striatal muscles, liver, and adipose tissue. In contrast, IGFs’ activity is responsible for long-lasting physiological processes determining cell fate such as cell energy metabolism, proliferation, differentiation, and inhibiting apoptosis [52]. The IGF-1 receptors have the highest expression in the cognitive brain, known for the highest energy demands, including the cortex, hippocampus, and thalamus [53,55,56]. Moderate expression is observed in olfactory bulbs, the hypothalamus, and the cerebellum. Lower expression is observed in the striatum, midbrain, and brainstem [55,56]. All these loci are of fundamental meaning for learning, memory, and homeostatic control.

Insulin receptor substrate 1 (IRS-1) is a signaling protein that plays a key role in transmitting signals from insulin and IGF-1 to intracellular pathways [51]. Importantly, the effects of IRS-1 are not limited to glucose metabolism. Its deficiency induces higher blood pressure likely due to lower vascular relaxation [51]. Peripherally, IRS governs osteoblast activity, thus coordinating bone turnover and metabolism. IGF resistance is observed under various physiological and pathological conditions, such as malnutrition and inflammation. It seems that IGF resistance may function as a safety valve inhibiting anabolic activities to sustain life [51]. It has been documented that reduced GH/IGF-1 signaling may prolong the life span [55].

Somatostatin regulates the endocrine system and also affects neurotransmission and cell proliferation [57,58,59]. It inhibits the secretion of the main factors controlling energy metabolism, i.e., insulin, glucagon, and growth hormone. Somatostatin release is triggered by low plasma pH and is inhibited by the vagus nerve. The hunger hormone (ghrelin) strongly stimulates somatostatin secretion, thus indirectly inhibiting insulin release. In the brain, somatostatin receptors are expressed in the arcuate nucleus, the hippocampus, and the brainstem nucleus of the solitary tract.

A growing body of evidence suggests that peripheral hormones released from the musculoskeletal system may control energy and also cognitive metabolism. One such example of such multi-organ interaction is the pleiotropic effects of the bone-derived hormone, osteocalcin. Osteocalcin is essential to regulate energy metabolism by increasing insulin sensitivity via the stimulation of adiponectin release by adipocytes [60]. The realized-by-bones osteocalcin increases also muscular glucose absorption and stimulates muscle secretion of IL-6, which again feeds back to bone cells to enhance osteocalcin production. Moreover, osteocalcin, along with other peripheral hormones, modulates brain activity involving spatial learning, memory, and even emotional states [61]. To date, it has been well documented the osteocalcin activity in the hippocampus and the ventral tegmental area of the midbrain where the hormone activity maintains the necessary level of cognitive metabolism (spatial learning and memory) whilst decreasing anxiety levels. Interestingly, the receptors of osteocalcin are also found in the somatosensory area, motor and auditory areas of the cortex, the piriform cortex, and the retrosplenial area, but their significance is not known yet. An intriguing scientific problem is whether the supplementation of osteocalcin can correct aging-related cognitive decline. Recent findings support this hypothesis indicating that uncarboxylated osteocalcin easily crosses the blood–brain barrier and thus may be used to correct cognitive deficits and decrease anxiety levels [61].

Such involvement of osteocalcin in addition to its progressive decline in midlife may suggest that the hormone is linked with age-related cognitive decline [61]. There has been found a correlation between a decrease in circulating osteocalcin levels and poor cognitive performance in healthy-aged individuals [61]. The levels of hormone change during the life span with a maximum during puberty when cognitive metabolism is the highest. After then, the circulating levels of osteocalcin decrease steeply before age 30 and age 45 in females and males, respectively [58]. Importantly, the levels of osteocalcin may be elevated by physical activity, which is known to improve cognitive function [61]. The release of osteocalcin upon exercise, however, is greatly diminished during aging [60].

## 4. Conscious Brain and Memory Traces

The essence of life is interaction and adaptation to the environment. To this end, an egocentric representation of the environment is created in the human brain which is constantly being attuned to any environmental changes. To this end, the functional neural networks formed in the cerebral cortex are continuously modified by the streams of sensory stimuli created during interaction with the environment [62,63]. Importantly, each sensory information is filtered through the individual emotional and motivational system, giving the reaction of the nervous network a subjective expression. In this way, the individual personality is built through the accumulation of all events and life experiences. Such a constituted personality determines the manner and intensity of subjective reactions to the subsequent, especially novel, life events. The dynamics of cognitive processes change throughout life depending on the amount and intensity of sensory stimuli that shape the level and intensity of cognitive metabolism. Additionally, cognitive metabolism depends on the sensitivity of the perceptual system that changes in the course of individual development. The aging of the brain especially impairs both cognitive and adaptive processes resulting in malfunctioning of local nerve structures and networks, thus fueling the vicious cycle of self-destruction in the brain.

Current views on brain consciousness are dominated by a holistic view, according to which specialized structures of the brain are assembled in neuronal networks which can be constantly modified by streams of sensory signals. Consequently, the interactions of the functional structures of the brain are established based on a dynamic equilibrium with the environment. In other words, the phenomenon of consciousness consists in selectively capturing relevant information from the environment and saving it in the form of memory traces. However, the output of egocentric control, particularly human motor behavior, seems more complex due to complex interactions between central (neural) and peripheral (executive) control.

Brain correlates of cognition are neuronal assemblies that during their entire lifetime are successively modified by perceptive streams. As a consequence, the inscribed in-brain-structure information creates an egocentric representation of ourselves and our surrounding environment that allows for conscious motor activity and social interactions modulated by emotional context. The interactions are sources of sensory signals stimulating brain functional networks. Brain–body–brain interactions are inseparable entities that are responsible for the regulation of both organismal homeostasis and allostasis with the latter being simply the anticipation of needs and preparing to satisfy them before they arise [64]. Importantly, the volume, multitude of neural connections, and thus longevity of brain correlates of cognition are determined directly by a process that can be called individual cognitive metabolism. This metabolism depends on the life experiences marked by the number and quality of environmental interactions, both material and social. Although the human brain consists mostly of single-life neurons, they have a unique capability to form and shape inherited neural networks in the processes of perception and learning. Brain neuroplasticity uses a variety of physiological processes allowing for changes in synaptic connectivity within neuronal networks that include forming new and eliminating unused synapses. The integral for creating brain networks, and for the overall architecture of brain connectivity, are synaptic processes from changes in their number and strength up to pruning unused ones. Moreover, the process of axonal myelination plays a fundamental role in the maturation and shaping of nervous networks, influencing both the speed of information transfer and the metabolic and energy efficiency of connections. Demyelinated, thus functionally or metabolically failing, axons are pruned, and neuronal cell bodies are finally directed to the path of apoptosis. Such pathophysiological sequence is a root cause of multiple sclerosis.

Cognitive changes in the brain use many different physiological processes with high energy consumption. The dominant role here is synaptogenesis (creating new synaptic connections) as well as pruning and removing unnecessary connections. Since neural connections are particularly loading for cognitive metabolism, their energy efficiency is improved by myelination of the axonal fibers. Both the creation of new permanent synaptic connections, their myelination and the removal of waste products are significant additional energy burdens for the already energy-consuming brain. Therefore, their course is spread over time and most often takes place during sleep. Thus, any sleep disturbance is a significant cause of cognitive metabolism dysfunctions and, consequently, is considered an early prodrome of dementia and neurodegenerative diseases.

Cognitive brain functions, including sensory perception and control of behavior, are ascribed to computation in networks of neurons [65]. Brain network plasticity is based on multiple physiological processes such as synaptogenesis, myelination, synaptic pruning, mitochondrial biogenesis, apoptosis, and limited adult neurogenesis. All neuroplastic processes can be activated during the entire life of the brain, though locally and with different dynamics [66]. They contribute to pronounced changes in cognitive abilities due to the optimization of the neuronal networks, more rapid neural communication, and integration of information across functionally related brain regions. The human neocortex is the primary candidate for the generation and maintenance of consciousness [67]. Human consciousness requires the uniquely human frontal lobe and its complex representation and control of limbic, thalamic, and brain stem regulatory systems [67].

It is commonly accepted that cognitive brain functions, including sensory perception, learning, motor control, and control of behavior, are ascribed to computation by axonal-dendritic chemical synapses in cortical networks [65]. Moreover, cognitive brain functions may occur either consciously or unconsciously. Neuronal substrates of human consciousness and cognition include firstly the neocortex, limbic and thalamic system, as well as the brain stem regulatory system [67]. The brain stem and midbrain regulation are essential to consciousness. The brain stem (phasic arousal) and midbrain (tonic activation) systems exert the limbic control of spatial (contextual) and object (focused) cognition, respectively [67]. The cytoarchitectonic features of limbic areas and their cortico-cortical pathways indicate high activity and plasticity, thus making them more susceptible to aging and neurodegeneration [16]. Indeed, pathological aggregation of aberrant proteins in Alzheimer’s and Parkinson’s diseases begins in limbic areas and spreads slowly to adjacent eulaminate areas [16]. It has been documented that during aging the activity of prefrontal limbic areas becomes less stable due to decreasing density of inhibitory parvalbumin neurons, which inhibit nearby pyramidal neurons [16]. In this context, dementia is the loss of memory and cognitive functions resulting from the aging-related disintegration of multiple interconnected functional networks.

The most numerous neurons found in the cerebral cortex, the hippocampus, and the amygdala, are multipolar pyramidal cells. The complex phenotype of pyramidal cells in the human prefrontal cortex imparts specific biophysical properties and connection patterns that differ from those in other cortical regions [68]. The ability of pyramidal neurons to integrate information depends on the number and distribution of the synaptic inputs they receive. A single pyramidal cell may receive about 30,000 excitatory (glutaminergic) inputs and 1700 inhibitory GABA inputs [69]. The functional nerve networks they create are continuously modified through the streams of sensory information that result from social and environmental interactions. These networks form the foundations of autobiographical memory.

The high functional plasticity of cortical networks in combination with raised energy supply allows for fast learning (i.e., modification of synaptic connectivity within all interacting networks) [63,70], but makes the pyramidal neurons prone to degeneration and aging. Maintaining cortical activity requires extraordinary energy inputs. It has been estimated that the brain cortex consumes roughly 34 mg/min of glucose of which the most, as much as 47%, is related to action potentials, and 13% is necessary to maintain the resting potentials of neurons [11,71].

It is commonly believed that neuronal energy metabolism could be assessed with the rate of oxidative phosphorylation. The total energy gained in the form of ATP per mole of glucose is around 1159 kJ, while the combustion of 1 mole of glucose releases 2870 kJ [72]. Consequently, the efficiency of oxidative phosphorylation in terms of energy obtained from glucose is relatively high, ranging up to 40% [72]. Additional energy is required for the maintenance of neuronal membrane resting potential which is based on the control of sodium, potassium, and calcium homeostasis. The ionic homeostasis, however, must be frequently perturbed to generate the action potentials that are transmitted, often over a long distance, to the synaptic junctions to release neurotransmitters. Synaptic transmission is vital for brain activity and gives value, strength, and meaning to nervous signals. Importantly, the greater the activity of neurons, the more disturbed their cellular homeostasis, and the more energy is required to recover it. Finally, synaptic transmission is the most energy-consuming process which determines the energy cost of cortical networks’ functioning and also their susceptibility to energy decline. This also underlines the crucial meaning of extracellular phosphate homeostasis for maintaining energy balance in a healthy brain.

The proper course of several physiological processes contributes to the exceptional plasticity of pyramidal neural networks. The primary mechanism of network formation is synaptogenesis, which changes the strength and quality of interneuronal connections [63]. This process is involved in the creation of both the excitatory glutamatergic synapses and the network of inhibitory (gabaergic) connections. Their range and interplay of the synaptic networks largely shape memory traces. An additional mechanism of shaping functional networks is the elimination or pruning of unnecessary synaptic connections. Another, not yet fully understood, mechanism of cortical plasticity is the process of axon myelination which allows optimization of the energy cost of axonal transmission [73]. When looking for the etiology of neurodegenerative changes, one should remember the adaptive trophic mechanism, important for the functioning and durability of nerve networks.

A substrate of memory trace within each neuronal network is a specific pattern of synaptic connections and gains, fixed and optimized in the process of learning. Moreover, the interaction of multiple functionally integrated cortical networks, storing various aspects of the given memory trace, ensures distributed and redundant, thus highly reliable, memory. Maintaining such an organized memory requires a lot of energy. Cortical nerve cells interacting within functional networks can create many thousand synaptic connections, and each synapse is the main consumer of energy obtained from oxygen and glucose. As much as 80% of energy consumption in neurons is attributed to synaptic connections [11]. The availability of energy could limit brain size and could determine a brain’s circuitry and activity patterns by favoring metabolically efficient wiring patterns. The brain’s energy requirements make it susceptible to damage during anoxia or ischemia, and understanding the demands made by different neural mechanisms may help the design of treatments.

A unique feature of the human brain is its cognitive abilities and the related ability to transform the structure of brain connections in response to any changes in the individual’s environment. Acquiring new skills and behaviors is defined as the learning process, the inseparable form of which is the creation of permanent memory traces. The set of memory traces accumulated during life determines our personality, the creation, and the maintenance of which, due to its size and dynamics, can be called cognitive metabolism. Cognitive metabolism is based on the modification (creating new and removing unnecessary) synaptic connections of the brain’s nerve networks, which represent the temporal-spatial characteristics of the environment as well as the motivational and emotional significance of the stimuli associated with them. Based on this information, patterns of motor responses are created. Thus, we have two components of cognitive metabolism that allow the transformation of streams of environmental stimuli into adequate motor responses. The dynamics of cognitive processes change in the course of individual development, reaching their maximum in the early developmental period and adolescence. In adulthood, these processes stabilize, and then, in the aging period, their intensity and scope gradually decrease. The manifestation of cognitive dysmetabolism is gradually increasing dementia, attention disorders, and a deficit of decision-making processes, as well as a deficit of short-term memory.

White matter accounts for more than half of the total brain volume. It is composed firstly of long axonal fibers with varying degrees of myelination [66,74] and other types of tissue including astrocytes and microglia. Glial fibrillary acidic protein (GFAP) is essential for normal white matter architecture and blood–brain barrier integrity, and its absence leads to late-onset dysmyelination [75]. Generally, the brain’s white matter plays a critical role in nearly every aspect of cognitive development, functioning, and brain involution [76]. Especially, myelinated pathways, which connect distributed brain areas, play a fundamental role in the maintenance of higher-level cognitive functions and are pivotal for cognitive, motor, and behavioral performance. Here, the process of myelination is decisive for the maintenance, modification, and overall efficiency of brain functioning. What is most important, myelin allows for limiting the neuronal transmission energy cost by the local distribution of mitochondria in the vicinity of the Ranvier nodes. Therefore, the processes of myelination and demyelination should be considered fundamental mechanisms of neuroplasticity. In ontogenesis, the process of myelination follows a complex topographical and temporal pattern that seems closely related to physiological development. During childhood, adolescence, and early adulthood, white matter maturation is fast and dynamic. The peak of myelination is observed at the age of 24 and 39 years. Next, in old age (70–80 years), the rate of myelination drops substantially to values comparable to 8-year-old values [76].

Brain development and its longevity are related to the pre-and post-natally accumulated energy metabolites and essential nutrients. At least two of these are recognized recently: nicotinamide riboside and sialic acid. Whereas the first one is directly related to neuronal energy metabolism, the second one is necessary for building and repairing cell membranes and myelin sheaths. Myelination and its plasticity are based on the reciprocal interactions of oligodendrocytes with the axons they ensheath. The integrity of axons depends on the glial supply of metabolites and neurotrophic factors. The relevance of this axoglial interaction can be observed in brain aging as well as in human myelin diseases. It has been well documented that myelination is dependent on the neuronal bioelectrical activity of the neurons and various molecular mediators synthesized and released in response to electrical events [77]. In particular, the brain networks can be modified in response to learning and new experience. Cognitive perception and learning are accompanied by increased neuronal activity which is the main driving force for dynamic myelination and shaping of the neuronal networks throughout the lifespan [77]. Axonal myelination increases neuronal energy efficiency and improves signal transmission by increasing axonal conduction velocity. Both factors seem to improve the longevity of neuronal circuits. Both factors contribute to phosphate metabolism which is the next determinant of brain physiology.

Adult neurogenesis and neuronal strongly contribute to the continuous remodeling of the neuronal networks emerging in cognitive processes [78]. Significantly, in the olfactory bulbs, hippocampus, and striatum, the incorporation and survival of new interneurons are activity-dependent. In the adult brain, neurogenesis in the subgranular zone of the hippocampal dentate gyrus and the subventricular zone of the lateral ventricle actively supplies neural stem cells and neural progenitor cells, respectively. During their migration through the rostral migratory stream, blood vessels form a scaffold for migration [79]. Importantly, in physiological conditions, limited apoptosis and hypoxia may induce neurogenesis and replacement of neurons in the adult cerebral cortex and other memory networks [78].

Neurotrophins play pivotal roles in the formation and plasticity of neuronal networks. In particular, the activity of brain-derived neurotrophic factor (BDNF) is vital to the survival, growth, and maintenance of neurons in key brain circuits involved in emotional and cognitive function [80]. Optimization of BDNF levels facilitates synaptic plasticity and remodeling, modulation of gene expression for plasticity, and resilience to neuronal insults [80]. In older adults, a deficiency of energy metabolism may result in chronic stress and depression. Stress-related decreases in BDNF levels as well as other neurotrophic factors could contribute to the atrophy of main limbic structures, including the hippocampus and prefrontal cortex which impairs cognition and awareness [80]. In parallel, the upregulation of BDNF occurs in the amygdala, performing a primary role in the processing of memory, decision-making, and emotional responses including fear, anxiety, and aggression [81]. A similar effect is observed in the nucleus accumbens which has a significant role in the cognitive processing of motivation, aversion, and incentive salience [81]. Deficits in incentive salience may contribute to avolition and anhedonia in the elderly, while the fearful forms of motivational salience may even contribute to some symptoms of paranoia. Additionally, BDNF abnormalities also contribute to the dysfunction of astrocytes and microglia which play a critical role in regulating neuroplasticity [80].

## 5. Pathomechanisms of Dementia

Dementia is a disorder that manifests as a set of related symptoms, which involve progressive impairments in memory, thinking, and behavior usually accompanied by emotional problems, difficulties with language, and decreased motivation, which altogether constitute human consciousness. In particular, the main symptoms of Alzheimer’s disease are cognitive malfunctions, particularly memory loss, impaired personality, and judgmental disorders. At least two pathogenic causes have been documented in the aging brain. The first is aging-related dysfunction of energy metabolism leading to neurodegeneration and massive death of neuronal cells. The second is a progressive involution of the brain in aging due to reduced cognitive and motor activity. Up to date, all attempts to treat dementias by removing their symptoms rather than their causes have failed.

Two fundamental factors are responsible for the aging and death of neurons that must be kept in mind when developing prevention strategies. Neurons are among the unique long-lived cells that have only one life. Throughout life, the number of neurons that make up the brain continues to decline, and the total number of synaptic connections changes. The second important factor is the optimization of energy expenditure, essential for life, by removing unnecessary or inefficient nerve cells. However, the ability to modify the structures of neural networks through cognitive processes remains throughout life. For the sake of simplicity, it can be assumed that the energy metabolism of the brain has two components: functional, related to the structure and maintenance of brain tissue, and cognitive, related to modification of the brain structure by learning and memory, emotions, and decision-making processes.

The energy crisis in the aging brain and the mass death of malfunctioning and unnecessary neurons impede the formation of new memory traces. The functional and trophic interaction inherent in the functioning of the brain has a particularly destructive effect on the functioning of the old and the formation of new functional networks in the elderly. This interaction ensures the correct energy supply and the necessary metabolites only of highly active neurons. It intensifies dysmetabolism, disturbs homeostasis, and accelerates the aging and death of nerve cells. The manifestation of these dysfunctions is late-onset neurodegenerative diseases. These diseases are preceded by prodromes signaling early metabolic brain dysfunctions such as arterial hypertension, type 2 diabetes, sleep disorders, anosmia, and mental changes with a characteristic increase in the level of anxiety. Later on, depending on the size and location of areas of the brain damaged, a set of quite specific symptoms appears. Since the effects of brain aging may appear rather late and usually concern different areas at the same time, the observed symptoms can be very diverse, which in the case of neurodegenerative diseases makes diagnosis and treatment extremely complicated. For instance, early clinical symptoms of Parkinson’s disease appear after 5–10 years after prodromes when almost 80% of dopaminergic neurons of the substantia niagra are dead [82].

Mitochondrial dysfunction has been implicated in the pathophysiology of cellular aging and neurodegeneration. Mitochondria signal stress by alterations in adenine nucleotide levels, reactive oxygen species(ROS) production, Ca^2+^ fluxes, permeability transition pore opening, and perhaps secretion of specific proteins/peptides [83]. Oxidative stress increases the number of malformed proteins that cannot be repaired due to a reduction in the synthesis of NAD-dependent enzymes. The rate of turnover in the cellular metabolic pathways is regulated based on reaction stoichiometry, the utilization rate of metabolites, and the translocation pace of molecules across the lipid bilayers. As a consequence of NAD deficiency in aging neurons, energy and metabolic crisis in the cells cumulate. When the energy deficit exceeds a critical level, the affected neurons are directed to the apoptotic pathway [84].

The correlation of pathological changes in the brain with accompanying symptoms allows for a better understanding of the pathophysiology of dementia. The most common feature of dementia is the loss of the ability to form new memory traces. This implies that brain aging and the resulting progressive degradation of cortical functional networks of the cerebral cortex and the hippocampus are the primary loci of dementia development. The neurodegenerative processes may be then spread to the limbic system, especially the amygdala as well as the entorhinal and cingulate cortex [85]. The also strikes neuronal systems controlling cortical arousal and metabolism. In Alzheimer’s disease, the adrenergic system of the locus coeruleus and the serotonergic neurons of the dorsal raphe are heavily affected [85]. The dorsal raphe nuclei play an important role in the sleep/wake cycle, which is disordered in the process of neurodegeneration. The raphe nuclei along with the locus coeruleus and the tuberomammillary nucleus project to the lateral hypothalamus. Their neurotransmitters, serotonin, norepinephrine, and histamine are fully active during waking hours, partially active during non-REM sleep, and have almost ceased during REM sleep [86].

Misfolded proteins have been widely considered to be the triggering factors in Alzheimer’s disease [87]. Indeed, the brains of patients contain large deposits of aggregated amyloid β-protein (Aβ) which is generated as a byproduct of the proteolytic processing of the amyloid precursor protein (APP). Cortical cognitive networks are formed in 90% of glutamatergic neurons. Importantly, a close relationship between APP processing, Aβ production, and activity has been found in glutamatergic neurons [88]. Synaptic activity within glutamatergic networks enhances Aβ release from nerve terminals, and, simultaneously, by activation of presynaptic group II metabotropic glutamate receptors, increases Aβ secretion [88]. The integrity of both processes allows for efficient control of Aβ metabolism in a healthy brain.

The APP is an integral membrane protein involved in synapse formation and neural plasticity. During cognitive processes, APP moves from the endoplasmic reticulum (ER) to the plasma membrane, where it is endocytosed by endosomes and lysosomes [89]. In physiological conditions, only a small fraction of the APP gives rise to Aβ [89]. Therefore initially, amounts of misfolded and oligomeric structures are fully controlled by astro- and micro-glia, and most of such waste products are removed by the glymphatic system during sleep. The initial steps of misfolding progress slowly until the minimum stable oligomeric unit is formed that then grows rapidly [87]. Both reduced cognitive activity and sleep disturbance in older adults cause the process is getting out of control, and the oligomeric structures to grow rapidly significantly disturbing brain homeostasis. Intensive generation and accumulation of Aβ lead to an excessive response of astrocytes releasing large amounts of proinflammatory cytokines that attract the microglia [89]. Additionally, reactive astrocytes aberrantly and abundantly produce the inhibitory gliotransmitter GABA. The released GABA inhibits neuronal activity which additionally disorders cognitive function. At this time, suppressing GABA production or release from reactive astrocytes fully restores neuronal activity, synaptic plasticity, learning, and memory [35]. However, the increasing size and intensity of misfolded protein contamination can make recovery ineffective.

Similarly, alpha-synuclein is a neuronal protein that normally regulates synaptic vesicle trafficking and subsequent neurotransmitter release [90]. It is found mainly in the axonal presynaptic terminals playing a core role in synaptic transmission, and consequently, alpha-synuclein is essential for cognitive processes. Its amount is closely correlated with neuronal activity, and its excess must be removed by glymphatic action. The protein misfolding and aggregation process appears to begin years or decades before the onset of clinical signs of dementia, suggesting that, to a certain extent, the waste product accumulation is tolerable and does not disturb significantly the cognitive function of the brain [90].

An increase in metabolic demand will lead to an increase in cerebral blood flow (CBF) [91]. The CBF and glucose metabolism remain closely coupled as they increase in proportion, whereas oxygen metabolism only increases to a minor degree through the so-called uncoupling of CBF and oxidative metabolism [91]. The trophic mechanisms involve both local angiogenesis as well as glia-neuronal metabolism-dedicated systems. Classically, the cognitive functions of the brain were considered to depend solely on dedicated elaborate networks communicating through synaptic connections and modulated by sensory feedback. This simplified view has been modified recently by the discovery of multiple neuro-glia interactions [92], and the existence of organismal endocrine cross-talk between the musculoskeletal and nervous systems have been found [60,93].

Brain activity and its energy metabolism are determined by specific, very complex interactions between neurons and glial cells. Particularly, astrocyte reactivity is triggered by any alteration in brain homeostasis [94,95,96]. Astrocytes, which are functionally coupled to neurons, are sensitive to local changes in levels of carbon dioxide, glucose, and sodium ions, which seem in turn to participate in controlling thirst and hunger. Astrocytes in complex metabolic interactions with neurons regulate the delivery of blood-borne metabolic substrates and firstly glucose, according to local neuronal energy needs. In the case of increased neuronal activity, more glucose is uptaken by the astrocytes and then, after fermentation, is supplied to neurons in the form of lactate [94,95,96].

The brain is considered to be autonomous in lipid synthesis with astrocytes producing lipids far more efficiently than neurons [96]. Astrocyte-derived lipids are taken up by neurons to support synapse formation and function [96]. Astrocytes are also the major source of lipid synthesis that provide neurons with cholesterol, essential for the formation of presynaptic vesicles used for transporting and release of neurotransmitters. Consequently, in aging astrocytes, the decline of cholesterol synthesis impairs neuronal communication within functional networks [96].

Neurons alone are unable to synthesize from glucose either the glutamate or the inhibitory GABA, and thus, they are fully dependent on astrocytes’ metabolic cycle of the glutamate/GABA-glutamine [94,97,98]. Functional changes occurring in senescence astrocytes include altered GABA/glutamate homeostasis, energy metabolism, and potassium homeostasis. Particularly, astrocytes deficiency of glutamate uptake from synaptic clefts leads to overstimulation of glutaminergic neurons culminating in their excitotoxic death. Microglia are also involved in regulating neuroinflammation by modulating immunological responses and playing a fundamental role in maintaining homeostatic brain functions. During development, microglia are involved in the pruning of unused synapses and clearance of apoptotic neurons [35,99].

Myelin, formed by oligodendrocytes, is a lipid-rich material that surrounds axonal fibers to insulate them and increases the rate and energetic efficiency of neuronal transmission. Besides that, the oligodendrocytes support axons metabolically by providing lactate as a nutrient [100]. The main component of myelin are gangliosides which account for 80% of all glycans and more than 75% of the sialic acid present in the brain [101]. A decrease in ganglioside levels and changes in the relative abundance of specific gangliosides is observed in the aging brain. Some studies have shown a direct role of ganglioside GD3 in apoptosis [101]. The GD3 induces apoptosis by its redistribution from the Golgi apparatus or plasma membrane to mitochondria. In the mitochondrial inner membrane, it triggers the opening of the permeability transition pores, playing a key role in oxidative phosphorylation. However, the excessive or uncontrolled opening of the transition pores may be harmful to the neurons by inducing apoptosis. Therefore, changes in the ganglioside profile are common in neurological conditions, including Alzheimer’s disease, Parkinson’s disease, Huntington’s disease, amyotrophic lateral sclerosis, stroke, multiple sclerosis, and epilepsy [101]. In the gray matter of the frontal and temporal cortex, gangliosides increase modestly from 20 to 50 years of age and then decreased slightly, with a more rapid decline at the age of 70 years. It is claimed that both rapid decline and excess of gangliosides result in severe neurodegenerative conditions [101]. Usually, the decrease of ganglioside levels is accompanied by similar changes in phospholipid and cholesterol which may indicate an aging-related increase in glial cells, loss of synapses, neuronal death, and brain tissue atrophy.

The myelination process is triggered and intensified in response to learning and new experiences leading to the formation of new memory traces. The myelinated fiber tracts integrate multiple brain areas associated with various cognitive, emotional, and motor functions. In young adulthood, when the individual level of accumulated experiences reaches its maximum, the white matter structure is relatively stable. Next, cognitive brain functioning and structure decline steadily. The common factor in older adults is decreased motor and cognitive activity that impacts neuronal brain networks. The process of demyelination strikes redundant and poorly functioning neuronal networks first.

The theory of retrogenesis claims that late-maturing brain tissue is particularly vulnerable during aging and that tissue degeneration in the aging brain follows the reverse sequence of tissue maturation [66,76]. According to this theory, cognitive development of the brain resembles constructing a pyramid, where an inherited brain structure serves as a base for additional layers added day by day. The top of the pyramid, which corresponds to the latest matured and myelinated network, is the most vulnerable to aging-related disorders [76]. It allows the claim that hypoactivity-related energy metabolism deficiency is a critical factor for cognitive decline associated with dementia. In Alzheimer’s disease and other forms of dementia, the hippocampus is the very first region of the brain to suffer damage [102]. On the other hand, aging-related reduced motor learning and hypoactivity seem to be responsible for the decay of motor-related brain networks and the development of Parkinson’s disease [102].

The regulation and coordination of complex cellular processes such as adult neurogenesis, cell migration, differentiation, and apoptosis depend on communication between neuronal and glial cells. The primary mediators of such physiological cell responses are receptor tyrosine kinases (RTKs). Additionally, insulin-like growth factor-1 (IGF-1) supports neuronal and glial differentiation, migration, and positioning in the brain [103]. Astrocytes play crucial roles in brain metabolism and are also involved in the neuroinflammatory response. They become reactive in response to virtually all pathological conditions such as synaptic damage, ischemia, infection, and neurodegenerative diseases [94]. In particular, the NF-κB signaling pathway is associated with neuroinflammation and apoptosis [94,104].

The brain is one of the greatest beneficiaries of energy processes and at the same time the most sensitive to energy deficit. Short-lasting energy deficiency usually causes a temporary decrease in motivation for physical activity, with increased signaling of energy nutritional needs. The changes are manifested in both motivation and mood. The problem of energy deficit begins to complicate the situation of elderly people, in whom the positive reinforcement systems (serotonergic and oxytocin) are malfunctioning, and fear and anxiety become dominant. The related stress disrupts the functioning of the whole organism, which consequently comes down to avoiding all activities and preferring a sedentary lifestyle. In such conditions, most of the cortical networks become poorly used and consequently deprived of energy. This opens the vicious circle of dementia. Decreasing metabolism in the aging brain combined with hypoactivity escalates muscle loss (sarcopenia) which in turn substantially reduces the capacity of striatal muscle to store and use glucose [104]. Additionally, aging-related hypoactivity increases visceral adiposity causing chronic systemic inflammation. The inflammation driven by hypoxia is considered a major cause of the development of obesity-associated diseases: insulin resistance, type 2 diabetes, and metabolic syndrome [104,105,106].

The first and most relevant for brain aging are energy metabolism-related changes, autonomic disturbances, deficient glucose metabolism (type II diabetes), and hypertension [26]. They are followed by sleep disorders, olfactory dysfunction, and psychiatric symptoms. These problems are related to intensive neurodegeneration of brain structures influencing sleep and wake states, circadian rhythm dysfunction, and motor and nonmotor symptoms [94,95].

## 6. Impact of Sleep Disorders on Glymphatic Waste Clearance and Dementia

The formation of memory networks and their modeling requires extraordinary energy inputs [107,108]. The neuroplastic processes are usually separated from daily brain activity and implemented during sleep. The quality of sleep decreases as we age, and disruption of the regular sleep structure is a frequent antecedent to the onset of dementia in neurodegenerative diseases [109,110]. Sleep has a restorative function by providing the clearance from the brain of metabolic waste products [110,111]. The efficiency of this process declines with age, suggesting a causal relationship between sleep disturbance and symptomatic progression in neurodegenerative dementias.

Sleep disorders negatively affect a wide range of cognitive functions, including memory, learning, attention, and emotional reactivity [112]. Consequently, chronic sleep deficits have emerged as a major risk factor for neurodegenerative diseases such as Alzheimer’s disease [112,113]. Multiple brain regions involved in sleep-wake control have been identified, including the brainstem, hypothalamus, and basal forebrain [114]. Sleep-active neurons have been found in several subregions of the hypothalamus [114]. Importantly, aging-related dysregulation of glucose metabolism results in elevated extracellular glucose levels which inhibit wake-promoting orexinergic neurons in the lateral hypothalamus and excite sleep-promoting GABAergic neurons in the ventrolateral preoptic nucleus [115]. This impact is destructive for the future cognitive functioning of the human brain and its metabolism and thus for the development of a broad spectrum of degenerative disorders. Additionally, the ventrolateral preoptic area is a key region promoting NREM sleep; the death of neurons involved in sleep function in the hypothalamic suprachiasmatic nucleus may explain sleep disturbance in aging [112].

Sleep has a critical function in ensuring metabolic homeostasis [111]. Sleep is characterized by reduced responsiveness to the external environment. The locus coeruleus is a noradrenergic center that sends broad projections to multiple brain areas and plays a pivotal role in regulating the transitions of vigilance states as well as a wide array of arousal-associated behaviors, such as attention, cognition, and orientation [116]. Physiologic sleep is associated with a 60% increase in the interstitial space, resulting in an increased exchange of cerebrospinal fluid with interstitial fluid and an increase in the rate of β-amyloid clearance during sleep [111]. The activity of the glymphatic system is increased during sleep when the clearance of harmful metabolites such as amyloid β increases two-fold relative to the waking state [117]. A fundamental tenet of brain homeostasis is that protein clearance must approximate protein synthesis [110]. Interstitial solutes, including protein waste, are removed through the glymphatic system and exported from the central nervous system via meningeal and cervical lymphatic vessels [110].

Sleep disorders cause an accumulation of waste products that additionally intensify brain dysmetabolism by impairing the functioning of the entire glymphatic system and, additionally, the process of adult neurogenesis which is another key mechanism in brain neuroplasticity and memory. Neurogenesis in the adult brain is limited to olfactory bulbs, hippocampus, and striatum, and the efficiency of the process relies on the migration of progenitor cells from the subventricular zone area via the rostral migratory stream to the target places [118]. Accumulation of toxic waste products limits the capacity of the migratory stream and vastly impairs adult neurogenesis.

The age-related impairment in sleep quality may cause the increasing impairment of glymphatic waste product clearance. Amyloid-β plaque formation is usually associated with an inflammatory response, including reactive micro- and astrogliosis [110]. Consequently, the age-related decline in CSF production, the decrease in perivascular AQP4 polarization, gliosis, and plaque formation all impede glymphatic flow and thereby impair waste clearance [110]. The sleep-wake cycle regulates fluid flow via the glymphatic system, setting the balance between protein clearance and aggregation. The increased incidence of aggregation-related disorders seen with aging depends on both vascular health and glymphatic patency [110]. Deterioration of the vascular bed leads to progressive demyelination and loss of white matter. A small vessel disease that targets the small cerebral vessels and results in the progressive thickening of their walls is common in hypertensive and diabetic patients [110]. Brain aging is associated with microangiopathy, suggesting that elderly individuals may have impaired metabolic activation of cerebral blood flow [119]. Hypoxia-ischemia brain injury is characterized by a pronounced inflammatory response, along with early structural alterations in the blood–brain barrier, followed by energy deprivation, the release of reactive oxygen species (ROS), and local inflammation [119].

Physiological sleep plays a vital role in the control of brain energy metabolism. Chronic sleep disorders can lead to impaired glucose metabolism [120]. After only a week of sleep restriction, subjects are unable to metabolize the glucose at rates observed in healthy, young individuals [120]. Sleep disorders disturb the endocrine regulation of energy homeostasis leading to weight gain and obesity [120]. A reduction of sleep duration to 4 h for two consecutive nights has been shown to increase ghrelin levels, as well as self-reported hunger [121]. After partial and total sleep deprivation, plasma cortisol level is significantly increased [122]. The most evident effect of cortisol activity is an increase in blood glucose concentration, further augmented by a decrease in the sensitivity of peripheral tissue to insulin. Cortisol also impacts the actions of hormones that increase glucose production, such as glucagon and adrenaline. Thus, sleep disturbances may induce chronic stress and accelerate the development of metabolic and cognitive disorders due to glucocorticoid excess [122]. Additionally, hypercortisolemia is deleterious to the bone by inhibiting osteoblast proliferation and bone formation, which may affect calcium/phosphate metabolism.

Poor sleep quality and circadian dysfunction can exacerbate neurodegeneration [123]. Sleep deprivation leads to cognitive deficits in domains including executive function, attention, and processing speed, as well as affective disturbances such as impulsivity and emotional lability [124]. Sleep deprivation has been found to induce neuroinflammation and aberrant protein homeostasis implicated in the neurodegenerative pathophysiology of Alzheimer’s disease [124]. Sleep alterations are a core component of mild cognitive impairment and Alzheimer’s Disease [109]. AD nighttime is characterized by a gradual decrease in slow-wave activity and a substantial reduction of REM sleep [116]. Sleep abnormalities can accelerate AD pathophysiology, promoting the accumulation of amyloid-β and phosphorylated tau [109]. The current concept is that sleep quality reduction results from amyloid-β aggregation may trigger hippocampal degeneration and, ultimately, memory impairment [109].

## 7. Dementia and Alzheimer’s Disease

Alzheimer’s disease is the most common form of dementia in the elderly [125]. Amyloid-β plaques and neurofibrillary tangles have been implicated in the disruption of neural function culminating in neuronal death. In patients with dementia, massive dying of neurons is observed in the hippocampus and cerebral cortex [125]. A relative increase of non-neuronal cell numbers in the cerebral cortex and subcortical white matter seems to suggest a causal contribution of reactive glial cell response to neuronal death [125].

Cortical atrophy dominates in dementia with Lewy bodies with an additional, concentrated area of atrophy in the subcortical brain, including the midbrain, hypothalamus, thalamus, basal ganglia, and substantia innominate [126,127]. It is characterized by fluctuating cognition, recurrent visual hallucinations, rapid eye movement sleep behavior disorder, and spontaneous parkinsonism [126,127]. The early event in the Alzheimer’s degenerative process is a major loss of synapses, dendritic complexity, and a decreased density of synaptic spines. The consecutive synaptic loss is most prevalent in the areas where neurodegenerative changes are observed. Massive synaptic dystrophy precedes the death of the cell body. The widely spread pathologies related to Lewy bodies and coexisting AD-type pathologies make the clinical manifestations complex and highly variable, increasing the difficulty of the differential diagnosis between dementia with Lewy bodies and AD. Remarkably, the substantia innominate is involved in the regulation of motivation and fear behaviors, the aggression-related emotional state. The nucleus basalis of Meynert plays a relevant role in cortical connectivity regarding memory, motor, and visual functions. Human studies have depicted its role in behavioral dysfunction such as aggressive behavior, impaired attention, perception, and memory involving visual discrimination and the processing of novelty and reward, as well as aversive signal processing. The substantia innominate functions imply that its atrophy might trigger psychiatric symptoms observed in patients with Lewy bodies dementia. A substantial neuron reduction in the nucleus basalis of Meynert in AD is responsible for the cholinergic deficiency in the cerebral cortex [128]. Many cholinergic neurons were shrunken rather than lost, suggesting retrograde degeneration in the nucleus basalis after the damage to the cortex [129]. The cholinergic system regulates various aspects of brain function, including sensory processing, attention, sleep, and arousal by modulating neural activity.

The pathological changes may appear locally or cover larger cortical areas. Growing evidence suggests that neurodegenerative diseases are caused by cortical network dysfunction rather than the dysregulation of an isolated brain region [12,13]. Local brain regions that are selectively damaged act as “nodes” in functional networks, representing the basis of the network degradation hypothesis [12]. According to this hypothesis, misfolded protein aggregates within small, selectively vulnerable neuron populations, may disconnect or weaken functional circuits. Consequently, the aberrant excitability disrupts neuronal homeostasis and function, leading to progressive degeneration of the entire functional networks [13]. Generally, the pattern of cortical gray matter loss in dementia shows thinning in the medial temporal cortex and frontal, parietal, and temporal neocortices, with relative sparing of the sensory, motor, and visual cortex. For instance, semantic dementia is associated with asymmetric, focal atrophy of the anterolateral temporal lobes. The medial temporal lobe, which has long been linked to memory function, appears to be similarly engaged in planning and anticipation of future events [13]. Frontotemporal lobar degeneration is a form of dementia affecting regions of autobiographical memory [13,14]. The core network includes medial temporal lobe structures such as the hippocampus and surrounding parahippocampal cortices, lateral temporal lobe cortices, posterior parietal regions including the posterior cingulate cortex and precuneus, as well as frontal regions such as the medial prefrontal cortex [14].

The long lifespan of neurons makes intracellular maintenance an important challenge [84]. Active transport along the axonal fibers is critical to neuron functioning. Deficits in retrograde axonal transport may cut off growth factor supply to long-range projecting neurons, resulting in axonal degeneration, synapse loss, and post-synaptic dendrite retraction [130]. Axonal transport supplies the distal synapse with newly synthesized proteins and lipids and clears damaged or misfolded proteins. The unique morphology of pyramidal neurons, highly polarized cells with extended axons and dendrites, makes them particularly dependent on active intracellular transport. Consequently, tau abnormalities have been linked to defective axonal transport in Alzheimer’s disease [85].

Neuropathologies and dementias are frequently linked with tau proteins that have become hyperphosphorylated insoluble aggregates called neurofibrillary tangles [131]. The tau protein is found in microtubules of axons where it provides structural support and serves to regulate the speed of the axonal transport. Without efficient axonal transport, axons undergo retrograde (Wallerian) degeneration which occurs in Alzheimer’s disease and other neurodegenerative conditions [84,131]. Failure to deliver, in time, sufficient quantities of the essential axonal protein, nicotinamide mononucleotide adenylyl transferases (NMNATs) is a key initiating event [130]. NMNATs are classically known for their enzymatic function of catalyzing NAD synthesis and thus gained a reputation as essential neuronal viability factors.

Vascular inflammation in the context of an altered blood–brain barrier (BBB) has been implicated in the pathogenesis of AD. Recent studies have shown that demented patients lose significant numbers of pericytes in the cortex and hippocampus leading to BBB damage, increased Aβ deposition, and tau pathology, culminating in severe cognitive impairment [132]. Astrocyte dysfunction also contributes to BBB breakdown and a decline in energy metabolism. Glucose uptake at the BBB, due to reduced GLUT1 transporter expression, is attenuated in patients with mild cognitive impairment, and this may escalate neurodegeneration and conversion to Alzheimer’s disease [132]. In addition, Aβ deposits in the vasculature enhance BBB pathological permeability in the aging brain. Additionally, the tau pathology also disrupts the integrity of the BBB, thus both tau and Aβ intensify BBB dysfunction, promoting neurodegeneration and cognitive impairment [132].

Microglia acts as the primary immune response in the central nervous system [35,133]. For this purpose, microglia constantly search the central nervous system for plaques, damaged neurons and synapses, and pathogens. The mobility of the microglia is particularly facilitated during sleep when the intercellular spaces are widened. They readily respond to even small changes in extracellular potassium levels, which usually signalizes demyelination or neuronal death. However, in the aging brain, the massive death of neurons may reach a critical level that makes microglia activity ineffective. Microglia form a protective barrier around amyloid deposits, compacting amyloid fibrils into a tightly packed and potentially less toxic form, thus reducing axonal dystrophy [134]. Triggering receptor expressed on myeloid cells 2 (TREM2) is required for microglial phagocytosis of a variety of substrates [134]. TREM2-deficient microglia fail to congregate or proliferate around plaques and show increased apoptosis [134,135].

Astrogliosis [135] and microgliosis [136] are common features of many neurodegenerative diseases with distinct etiologies [134,137,138,139,140]. In the CNS, microglia serve as resident phagocytes that dynamically survey the environment, playing crucial roles in brain tissue maintenance, injury response, and pathogen defense [134,141,142]. An important aspect of brain tissue homeostasis is microglial phagocytosis of a variety of substrates, including apoptotic neurons, LDL and other lipoproteins, and Aβ and clearance of debris [134]. Thus, in physiological balance, the microglia protect against the incidence of dementia. However, there is also a growing body of evidence for excessively activated microglia being harmful to neurons and mediating synapse loss by engulfment of synapses [134]. In response to the brain injury or chronic disorder, activated microglia congregate around degenerating neurons, and may produce toxins and inflammatory cytokines that escalate the process of neurodegeneration [142,143,144]. The sustained exposure of neurons to pro-inflammatory mediators can cause neuronal dysfunction and contribute to cell death [34,35,133].In particular, systemic inflammation may drive the pathogenesis of neurodegenerative diseases by augmenting neuroinflammation [133]. In the process of neurodegeneration, the massive death of neurons reaches the critical point where microglia is unable to cope with the excess necrotic neurons and toxic proteins which terminally end the energy metabolism of the brain. The problem of an uncontrolled massive death of neurons strikes firstly the cortical networks leading to the clinical phase of Alzheimer’s disease.

## 8. Concluding Remarks

The increasing population of Europe coupled with the aging demographics in many European countries provides a clear indication that the overall number of people with dementia is likely to continue to increase significantly. Up to date all attempts to treat neurodegenerative diseases by removing their symptoms rather than their causes have failed. Therefore, in the presented narrative review, I attempt to explain the etiology of dementia and Alzheimer’s disease from the perspective of energy and cognitive metabolism dysfunction in an aging brain. I hope that this perspective, though perhaps too simplified, will bring us closer to the essence of neurodegenerative disorders and will soon allow us to develop new preventive/therapeutic strategies in our struggle with dementia.

The knowledge gathered so far allows us to posit that the basic cause of brain aging is the loss of control over glucose metabolism, resulting in the loss of energy balance. Homeostasis is ensured by balancing the processes of glucose intake and accumulation in skeletal muscles, the liver, and adipose tissue, and then their activity-dependent redistribution within the organism. While glucose intake and storage is a short-term process controlled by insulin, the use of this energy fuel is controlled by the growth hormone/IGF-1 cycle, whose long-term activity allows for the continuation of the cognitive brain activity during sleep. Three key factors responsible for brain aging and dementia are (1) the loss of the brain’s ability to sense blood glucose levels, (2) decreased IGF1 activity and (3) sleep disorders. The implemented-by-brain compensatory processes, including type 2 diabetes and hypertension, occur to be more pathogenic. Increased glucose levels that cannot be utilized by the brain and muscles result in hypoactivity which dramatically restricts cognitive processes. The unused neuronal networks are systematically reduced, which triggers the development of dementia. Moreover, almost all the most energy-consuming nervous structures are reduced. Besides cortical pyramidal neurons, age-related neurodegeneration also strikes the brain pyramidal pathways including the nigrostriatal control system and other long and unmyelinated axons of the autonomic nervous system.

The glucose/energy crisis also impacts emotional control. Chronic stress accompanying energy dysmetabolism potentiates sleep disorders and additionally triggers the process of gluconeogenesis contributing strongly to hypoactivity-dependent sarcopenia. Simultaneously, sleep disorders devastate the functioning of the glymphatic system with its two major physiological functions: waste product removal and adult neurogenesis. Taken together, all pathophysiological and behavioral processes cumulate, progressively reaching the critical point of microglia deficiency and uncontrolled neurodegeneration. This is the final act of conscious brain self-destruction. If it is the case, how we can help people suffering from dementia?

Massive death of cortical neurons ultimately terminates the functioning of the cognitive brain. The dead neurons will never recover, but early intervention may slow down or even reverse the cascade of neurodegeneration. Intensified cognitive metabolism supplemented by essential energy nutrients such as NAD, IGF-1, and probably vitamin D and phosphate should improve the functioning of the cortical networks. If we combine it with a low-fat diet and physical activity, we can anticipate better functioning and maintenance of the cognitive networks in good shape until a late age. This opens a window of opportunity to treat or slow down the process of dementia before it reaches a critical point with no return.

In this context, it is worth mentioning the NAD and phosphate contribution to brain aging and neurodegeneration. There is rapidly growing research on the role of vitamin B3 precursors (e.g., nicotinamide riboside chloride) in supporting neuronal viability. In the case of senile neurons, the curing properties may occur rather limited. Supplementation of vitamin B3 precursors may, however, slow down the aging process in cortical neurons and their cognitive networks. At the same time, it should be remembered that only neuronal cells cognitively stimulated show increased activity and may finally benefit from NAD supplementation and improved energy metabolism. Similarly, there is an underrated problem of phosphate/calcium (Ca/P) homeostasis and its impact on brain energy metabolism. An increasing body of evidence suggests that the physiology of the brain depends strongly on bone-controlled calcium and phosphate homeostasis. Age-related dysfunction of Ca/P homeostasis strongly impacts the functioning of neural and muscular cells. The dysfunction is initially observed as a fatigue syndrome. Phosphate plays an important role in brain energy metabolism. Its inflow to neurons allows for glucose phosphorylation which is the initial step of oxidative phosphorylation and ATP production. A combination of hypophosphatemia and elevated glucose level may be devastating for elderly subjects. Therefore, recovery of Ca/P homeostasis is the second main target in the treatment of neurodegenerative diseases. Recent studies on the pathogenic effect of hypophosphatemia suggest that vitamin D supplementation may improve energy metabolism while removing symptoms of chronic fatigue. This gives new hope to elderly people suffering from neurodegenerative diseases.

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
