# Peer review of "Pathogenesis of Dementia"

_ijms, 2022, doi:10.3390/ijms24010543_

Round 1
Reviewer 1 Report
In the manuscript entitled “Pathogenesis of dementia”, Dr. Blaszczyk reviews the etiology of dementia and Alzheimer's disease from the perspective of energy and cognitive metabolism dysfunction in an aging brain.
Overall, this is a good and timely article summarizing several sets of the novel and highly relevant results and proposing a new diagnostic approach.
However, in its current form, this work has several minor issues that could be addressed.
· In chapter #5 (lines 591-593) Pathomechanisms of dementia, the author suggests that “In particular, the main symptoms of Alzheimer's disease are cognitive disorders, particularly memory loss, impaired personality, and judgmental disorders.” The term “disorders” is misleading in this context. I propose to change it to “malfunctions”.
· In chapter #7 (line 909) the author suggests that “Alzheimer’s disease is the most common form of dementia in the elderly.” However, there is no reference provided to support it.
· The author correctly indicates that poorly controlled blood sugar may increase the risk of developing dementia. Due to this relationship some have called Alzheimer’s “diabetes of the brain” or “type 3 diabetes (T3D)”. The author could cite some of the recent papers, which explore this phenomenon. For example, this one (1).
· Recent evidence suggests that AD is characterized by distinctive abnormalities apparent on systemic, histological, macromolecular, and biochemical levels. Interestingly, several findings demonstrate that arginine metabolism is dramatically altered in diverse regions of AD brains (2).
· The author pointed in several places to metabolic abnormalities associated with the development of dementia. Some authors apply a broader approach and categorize Alzheimer’s disease as a brain expression of a systemic complex metabolic disorder, which shares similarities and pathogenic pathways with diabetes mellitus, obesity, and atherosclerosis (3).
References:
1. Nguyen TT, Ta QTH, Nguyen TKO, Nguyen TTD, Giau VV. Type 3 Diabetes and Its Role Implications in Alzheimer's Disease. Int J Mol Sci. 2020 Apr 30;21(9):3165. doi: 10.3390/ijms21093165. PMID: 32365816; PMCID: PMC7246646.
2. Liu P, Fleete MS, Jing Y, Collie ND, Curtis MA, Waldvogel HJ, Faull RL, Abraham WC, Zhang H. Altered arginine metabolism in Alzheimer's disease brains. Neurobiol Aging. 2014 Sep;35(9):1992-2003. doi: 10.1016/j.neurobiolaging.2014.03.013. Epub 2014 Mar 20. PMID: 24746363.
3. Polis B, Samson AO. A New Perspective on Alzheimer’s Disease as a Brain Expression of a Complex Metabolic Disorder. In: Wisniewski T, editor. Alzheimer’s Disease [Internet]. Brisbane (AU): Codon Publications; 2019 Dec 20. Chapter 1. Available from: https://www.ncbi.nlm.nih.gov/books/NBK552149/ doi: 10.15586/alzheimersdisease.2019.ch1
Author Response
Dear Editor,
I would like to thank the Reviewers for their suggestions and comments which allowed me to improve my Review. I corrected my text approving most if not all their comments. My detailed responses are shown below. I think that the present version of my article is suitable for publication in your prestigious journal.
Reviewer stated: this work has several minor issues that could be addressed.
- In chapter #5 (lines 591-593) Pathomechanisms of dementia, the author suggests that “In particular, the main symptoms of Alzheimer's disease are cognitive disorders, particularly memory loss, impaired personality, and judgmental disorders.” The term “disorders” is misleading in this context. I propose to change it to “malfunctions”.
- In chapter #7 (line 909) the author suggests that “Alzheimer’s disease is the most common form of dementia in the elderly.” However, there is no reference provided to support it.
- The author correctly indicates that poorly controlled blood sugar may increase the risk of developing dementia. Due to this relationship some have called Alzheimer’s “diabetes of the brain” or “type 3 diabetes (T3D)”.The author could cite some of the recent papers, which explore this phenomenon. For example, this one (1).
- Recent evidence suggests that AD is characterized by distinctive abnormalities apparent on systemic, histological, macromolecular, and biochemical levels. Interestingly, several findings demonstrate that arginine metabolism is dramatically altered in diverse regions of AD brains (2).
- The author pointed in several places to metabolic abnormalities associated with the development of dementia. Some authors apply a broader approach and categorize Alzheimer’s disease as a brain expression of a systemic complex metabolic disorder, which shares similarities and pathogenic pathways with diabetes mellitus, obesity, and atherosclerosis (3).
References:
- Nguyen TT, Ta QTH, Nguyen TKO, Nguyen TTD, Giau VV. Type 3 Diabetes and Its Role Implications in Alzheimer's Disease. Int J Mol Sci. 2020 Apr 30;21(9):3165. doi: 10.3390/ijms21093165. PMID: 32365816; PMCID: PMC7246646.
- Liu P, Fleete MS, Jing Y, Collie ND, Curtis MA, Waldvogel HJ, Faull RL, Abraham WC, Zhang H. Altered arginine metabolism in Alzheimer's disease brains. Neurobiol Aging. 2014 Sep;35(9):1992-2003. doi: 10.1016/j.neurobiolaging.2014.03.013. Epub 2014 Mar 20. PMID: 24746363.
- Polis B, Samson AO. A New Perspective on Alzheimer’s Disease as a Brain Expression of a Complex Metabolic Disorder. In: Wisniewski T, editor. Alzheimer’s Disease [Internet]. Brisbane (AU): Codon Publications; 2019 Dec 20. Chapter 1. Available from: https://www.ncbi.nlm.nih.gov/books/NBK552149/ doi: 10.15586/alzheimersdisease.2019.ch1
Author Response: I am grateful for the insightful analysis of my Review. I have modified my text according to all suggestions of Reviewer # 1. I have also added 3 suggested references.
Reviewer 2 Report
The manuscript submitted by Blaszczyk sets out to describe the etiology of dementia and Alzheimer's specifically from the perspective of energy and metabolism as they relate to cognition. The manuscript does a thorough job including many aspects of cognition-related energy and metabolism, making for a comprehensive resource on the topic. The primary weakness is organization/structure of the manuscript, as detailed below.
General Comments:
As noted above, the author does an excellent job including a breadth of information as it pertains to energy and metabolism associated with cognition. The thorough descriptions of oxidation and respiration biochemistry, energy consumption of specific brain structures and cells, and how these processes go awry are greatly appreciated. Gaps in understanding are highlighted.
Much of the important content is included; however, there is significant redundancy throughout the manuscript with little to no new information each time a concept is introduced. One significant omission is the topic of microglia. They are briefly mentioned in sections 1-4, but its not until section 5 (Pathomechanisms of dementia) where they are covered more fully. Yet even within that section, important information about their own metabolism and dysfunction is missing.
Overall, the manuscript is well structured, but would benefit from additional sub-headings to help organize content. Furthermore, the additional structure with perhaps some reorganization would make the manuscript more concise, increasing readability and creating space for the author to make novel and insightful conclusions about the studies referenced, or the field in general. Due to the current manuscript length, it may also benefit from focusing on just Alzheimer’s disease, rather than trying to include PD/synucleinopathies as well.
The cited references are a bit out of date with 85 of them (59%) being at least 5 years old. While some older references are expected to provide historical context and cite original findings, an emphasis on more recent studies would make the manuscript more relevant and up to date. Furthermore, at least 60 (42%) are themselves review articles. In a review such as this with 144 references, one would expect there to be more reliance on primary research articles. There is an excessive number of self-citations.
There are no figures/tables/images/schemes, and the manuscript would benefit greatly from their inclusion. Specifically, schematics for the various metabolic processes and their connections and relationships, as well as a summary figure/schematic for how they contribute to dementia, would significantly enhance the manuscript and the reader’s understanding of the text.
The statements and conclusions drawn from the listed citations are coherent; however, aside from recapping previous findings and identifying the gaps for future work, the author does not make any novel or insightful conclusions about the studies referenced, or the field in general.
The quality of the writing and English is good, but would benefit from some attention. Please be sure all abbreviations are spelled out at the first use. There are also multiple places where there appear to typographical and/or grammatical errors that at times make the author’s meaning unclear.
Specific Comments
Lines 106-131 cover NAD, however there is no mention of microglia. Their own energy usage/metabolism is important with respect to their functions related to neurotrophic support.
Starting with section 3 (Systemic dysmetabolism), specific brain structures are italicized. This wasn’t the case prior, and it continues with some irregularity throughout the rest of the manuscript. The significance of italicizing these words are not clear.
Lines 297-289 mention IGF-1 dysregulation in disease states, but don’t provide any mechanisms. Please elaborate on this point as it otherwise is left hanging.
Please avoid repetitive usage of words like “thus” which appears multiple times in lines 367-375.
The meaning of lines 392-394 is unclear. Please revise.
Lines 409-423 seems disconnected from earlier sections, feeling disjointed and random. Including some context/framework would be appreciated.
Lines 424-445 is another specific place that would benefit from a summary schematic.
Deficiency of the glymphatic system is mentioned in line 567, coming from nowhere. Please provide background information so the reader understands the significance of these findings.
Within section 5, microglia are portrayed as responding to changes in neurons and astrocytes when in fact they have surface receptors that recognize amyloid species, stimulating their activation and inflammatory responses. Please revise to include these additional factors.
Author Response
Dear Editor,
I would like to thank the Reviewers for their suggestions and comments which allowed me to improve my Review. I corrected my text approving most if not all their comments. My detailed responses are shown below. I think that the present version of my article is suitable for publication in your prestigious journal.
Reviewer # 2
Criticism # 1: There is significant redundancy throughout the manuscript with little to no new information each time a concept is introduced. One significant omission is the topic of microglia. They are briefly mentioned in sections 1-4, but its not until section 5 (Pathomechanisms of dementia) where they are covered more fully. Yet even within that section, important information about their own metabolism and dysfunction is missing.
Author response: I have restructured the text by adding new subtitles. In the present version, I have divided the text of the Introduction and the subtitles will guide the reader to different aspects of energy dysmetabolism and neurodegeneration. I am sorry for some redundancy, but I wanted to give a reader a chance to find the almost complete knowledge of the problem within each chapter without the necessity to return to previous chapters.
Criticism # 2: Overall, the manuscript is well structured, but would benefit from additional sub-headings to help organize content. Furthermore, the additional structure with perhaps some reorganization would make the manuscript more concise, increasing readability and creating space for the author to make novel and insightful conclusions about the studies referenced, or the field in general. Due to the current manuscript length, it may also benefit from focusing on just Alzheimer’s disease, rather than trying to include PD/synucleinopathies as well.
Author response: I have done it on purpose. I aimed to show the reader that the problem of dementia is closely related to natural brain aging and neurodegeneration. Whereas dementia impacts cognitive memory, the PD impairs the motor memeory, but the root cause on a cellular level is the same.
Criticism # 3: The cited references are a bit out of date with 85 of them (59%) being at least 5 years old. While some older references are expected to provide historical context and cite original findings, an emphasis on more recent studies would make the manuscript more relevant and up to date. Furthermore, at least 60 (42%) are themselves review articles. In a review such as this with 144 references, one would expect there to be more reliance on primary research articles. There is an excessive number of self-citations.
Author response: I understand the reviewer's reservations, but this is a serious problem that affects all of modern science. According to the PubMed database, more than 16,500 scientific articles on Alzheimer's and dementia have been published in the last twenty years alone. Most of them repeat the same messages. Many authors focus on single mechanisms or even genes in dementia research. In my review, I wanted to present a new perspective on brain aging and neurodegeneration based on changes in energy metabolism. Therefore, I have cited several of my papers where I discuss in detail the various mechanisms of neurodegenerative changes in Parkinson's disease that have much to do with dementia. I apologize for the excess of my work, but this has nothing to do with self-promotion. Moreover, I deliberately cited many of up-to-date review papers to avoid getting lost in the details.
Criticism # 4: There are no figures/tables/images/schemes, and the manuscript would benefit greatly from their inclusion. Specifically, schematics for the various metabolic processes and their connections and relationships, as well as a summary figure/schematic for how they contribute to dementia, would significantly enhance the manuscript and the reader’s understanding of the text.
Author response: I'm very sorry, but I'm not a good graphic artist, and drawing diagrams or figures is difficult for me. Besides, after 40 years of work in neurophysiology, I find that each of the mechanisms of functioning of the brain and neurons, in particular, is so complicated and interconnected with other tissues, e.g. with muscles and bones, that any graphical representation may be too far-reaching simplification.
Criticism # 5: The statements and conclusions drawn from the listed citations are coherent; however, aside from recapping previous findings and identifying the gaps for future work, the author does not make any novel or insightful conclusions about the studies referenced, or the field in general.
Author response: I have included my suggestions for future research in the Concluding remarks. I am convinced that work on neuronal energy metabolism will be beneficial to the elderly and patients with dementia.
Criticism # 6: The quality of the writing and English is good, but would benefit from some attention. Please be sure all abbreviations are spelled out at the first use. There are also multiple places where there appear to typographical and/or grammatical errors that at times make the author’s meaning unclear.
Author response: Sorry for my Polish version of American English. I have checked the text carefully and corrected the mistakes.
Reviewer’s Specific Comments:
- Lines 106-131 cover NAD, however there is no mention of microglia. Their own energy usage/metabolism is important with respect to their functions related to neurotrophic support.
Author response: I am not aware of any publication on NAD in microglia. The issue of NAD deficiency in the aging brain is quite new and, according to me, there is no study on microglia energy metabolism.
- Starting with section 3 (Systemic dysmetabolism), specific brain structures are italicized. This wasn’t the case prior, and it continues with some irregularity throughout the rest of the manuscript. The significance of italicizing these words are not clear.
Author response: I have corrected the problem of italicized names of brain structures according to suggestions.
- Lines 297-289 mention IGF-1 dysregulation in disease states, but don’t provide any mechanisms. Please elaborate on this point as it otherwise is left hanging.
Author response: Here I have only mentioned the problem of IGF. I have returned to it in the next paragraphs. It is a very important issue since it seems that the activity of IGF1 is more fundamental for brain functioning than insulin. I showed in my review that in physiological conditions, the insulin could be active for about 10 minutes, but IGF sharing the same receptor with insulin is active for several hours. Type 2 diabetes causes prolonged insulin production becomes chronic dysfunction leading to insulin resistance. The IGF activity is the next direction of research.
- Please avoid repetitive usage of words like “thus” which appears multiple times in lines 367-375.
Author response: Redundant “thus” has been removed from the text.
- The meaning of lines 392-394 is unclear. Please revise.
Author response: The lines have been removed from the text.
- Lines 409-423 seems disconnected from earlier sections, feeling disjointed and random. Including some context/framework would be appreciated.
Author response: The lines have been removed from the text.
- Lines 424-445 is another specific place that would benefit from a summary schematic.
Author response: I am sorry but I am not able to draw nice schematics.
- Deficiency of the glymphatic system is mentioned in line 567, coming from nowhere. Please provide background information so the reader understands the significance of these findings. Within section 5, microglia are portrayed as responding to changes in neurons and astrocytes when in fact they have surface receptors that recognize amyloid species, stimulating their activation and inflammatory responses. Please revise to include these additional factors.
Author response: I have added references for the glymphatic system functioning. In the section on the impact of sleep disorders, I explained the details of waste product removal by the glymphatic system. I have also explained the problem of adult neurogenesis deficiency due to age-related glymphatic system impairments. For readers interested in this problem I referred them to my publication on PD, Acta Neurobiol. Exp. 2016.
Reviewer 3 Report
In this article by Janusz W. Błaszczyk, the author describes the pathogenesis of dementia. Similar review has already been published by Yin et. al. (PMID: 27154981). There is no significance or novelty if we compare with the exiting review. I will recommend the author to rewrite the article and make subsections so that reader can easily understand the full story.
Author Response
I have revised my manuscript pointing toward the root causes of dementia. I have also added references to Yin et al. 2016. My review focuses on quite different aspects of the causes of dementia and points to potential methods to prevent or treatment of neurodegenerative diseases. As you compare both texts, they are completely different and I do not understand why you reject my review without comments.
Round 2
Reviewer 2 Report
The author's revisions to the manuscript are greatly appreciated and have resulted in a very nice review.
Author Response
Thank you very much.
Reviewer 3 Report
1. Author improves a lot in revised version, for example, addition of subsection, grammatical errors. However, author previously did not cite Yin F et al which was the major concern of rejection.
2. At least 2 figures are still needed under section 2 and 3 which will improve the manuscript as well as readers better understanding.
3. It would be better to focus on Alzheimer's disease not Parkinson's disease as the title is "Pathogenesis of dementia"
Author Response
Dear Reviewer
Thank you very much for your suggestions. I have added a reference to Yin et al. 2016. In the revised version it is ref # [150]. I have a problem with the figures since the root causes of dementia are so complex and intertwined that every schematic figure rather complicates the explanation. Moreover, there are several detailed figures in Yin et al. review.
Finally, I cannot agree with you on the problem of Parkinson's disease. Simply in my review, I wanted to underly that aging-related mechanisms of neurodegeneration are common for both cognitive memory (dementia) and motor memory (Parkinson's disease. The only difference is in the locus. The first one is mostly a problem with the neocortex while in the second one there is a problem with the basal ganglia. This position may be helpful for clinicians to develop a treatment strategy. Removing these few sentences would not change much but I am convinced that motor-cognitive interaction has fundamental meaning for dementia development. So please let me publish my text as is.